# Deep Smoothing of the Implied Volatility Surface

**Damien Ackerer**
UBS
Zürich, Switzerland
damien.ackerer@epfl.ch

**Natasa Tagasovska**
Swiss Data Science Center
Lausanne, Switzerland
natasa.tagasovska@sdsc.ch

**Thibault Vatter**
Department of Statistics
Columbia University
New York, USA
thibault.vatter@columbia.edu

## Abstract

We present a neural network (NN) approach to fit and predict implied volatility surfaces (IVSs). Atypically to standard NN applications, financial industry practitioners use such models equally to replicate market prices and to value other financial instruments. In other words, low training losses are as important as generalization capabilities. Importantly, IVS models need to generate realistic arbitrage-free option prices, meaning that no portfolio can lead to risk-free profits. We propose an approach guaranteeing the absence of arbitrage opportunities by penalizing the loss using soft constraints. Furthermore, our method can be combined with standard IVS models in quantitative finance, thus providing a NN-based correction when such models fail at replicating observed market prices. This lets practitioners use our approach as a plug-in on top of classical methods. Empirical results show that this approach is particularly useful when only sparse or erroneous data are available. We also quantify the uncertainty of the model predictions in regions with few or no observations. We further explore how deeper NNs improve over shallower ones, as well as other properties of the network architecture. We benchmark our method against standard IVS models. By evaluating our method on both training sets, and testing sets, namely, we highlight both their capacity to reproduce observed prices and predict new ones.

## 1 Introduction

The implied volatility surface (IVS) is a key input for computing margin requirements for brokers, quotes for market makers, prices of exotic derivatives for quants, and strategies positions for traders. As a result, tiny predictions errors can lead to dramatic financial losses. But standard IVS models often lack the ability to flexibly reproduce market prices and value other instruments without quotes. In this paper, we merge known ideas from different fields, namely machine learning (ML) and mathematical finance, to build a new solution for a non-trivial and relevant financial problem: the interpolation and extrapolation of the IVS. More specifically, we use a neural network (NN) to correct the IVS produced by any standard model. This lets practitioners plug our method on top of existing approaches while offering an unprecedented trade-off between flexibility and computational complexity. This problem was initially brought to us by a financial institution willing to improve its existing solution and a version of this method is being tested in production by a financial institution, where thousands of models are continuously updated throughout the day and used as inputs to various quantitative tools.

Practitioners have a growing interest to leverage NNs as flexible predictors for applications requiring an understanding of the complex dynamics of financial markets. And the ML community continues

to build better tools and understanding deep models. But leveraging domain expertise about a specific problem is often difficult, and still an active field of research. In this paper, we show how knowledge from both ML and mathematical finance can be merged to build a well performing and consistent hybrid model. Our application focuses on options, yet, the same approach could be similarly applied to other financial problems.

**Problem setting and background context.** An *option* is a financial contract giving the option holder the right to buy (a call option) or the right to sell (a put option) an asset, such as a stock or a commodity, for a predetermined price (the *strike price*) on a predetermined date (the *expiry* date). An initial *premium* must be paid to the option seller in order to acquire today the right to buy or sell and asset in the future at, possibly, a preferential price. The standard, textbook approach to model option pricing is based on the so-called Black-Scholes (BS) formula. The Black-Scholes formula provides a closed-form formula for option prices for a specific stock price model, the geometric Brownian motion (GBM). However, the formula builds on unrealistic assumptions such as continuous price trajectory and trading, absence of market frictions such as bid-ask spread and integer contract size, and normality of log-returns. Options are traded on financial exchanges, and their prices typically invalidate the GBM model. The Black-Scholes formula is a convenient one-to-one mapping between a price and a volatility parameter that is preferred by practitioners for a variety of reasons. For example, it allows them to easily compare the price of options with different contract strike prices and maturities. As a consequence, a lot of domain expertise has been developed for the so-called *implied volatility surface (IVS)*: the continuous representation of this volatility parameter expressed as a function of the strike price and of the expiry at a given point in time. However, only a finite number of option prices are observed in practice. In addition, quoted option prices should not allow *arbitrage opportunities*, that is, constructing a portfolio that may generate profits at a zero initial cost. Therefore the two main challenges when modeling the implied volatility surface are, first, to ensure that the corresponding option prices do not allow arbitrage opportunities and, second, the generalization to an entire surface given a limited number of observations. Such a construction allows the IVS to be used as input to price financial derivatives in a consistent way, and that enables effective risk-management. It is then worth noting that some commonly used models, such as SABR [26] and SVI [18], may not be arbitrage-free for some parameters values, see for example [48, Section 3], which contradicts real-life scenarios.

**Short literature review.** As neural networks and machine learning in general, prevail almost all aspects in science and industry, finance applications have also been impacted [30, 21, 31, 9, 29]. Deep learning have been studied with applications to option pricing in [49, 27, 17, 6, 37, 42, 43]. These papers exploit the well-known universal approximation property of neural networks [35, 34, 41]. Several applications of NN models for implied volatility smoothing exist, see [14, 53, 44, 38, 52] with more details in Appendix A. [53] in his PhD thesis also studied the use of soft constraints to train an arbitrage-free model. Extensive domain specific knowledge has been built on the IVS. In this work, we will use the no-arbitrage constraints on the IVS derived in [48], and later refined in [24], the moment condition of [40], and the surface SVI (SSVI) model of [20]. A more exhaustive literature review is available in Appendix A.1.

**Summary of contributions.** We present a new methodology to correct, interpolate, and extrapolate the implied volatility surface in an arbitrage-free way. We achieve this by modeling the implied total variance as a product of a neural network and a prior model, and by penalizing the loss using soft constraints during training so as to prevent arbitrage opportunities. The prior model should be a standard valid model for the total variance that will guide the general shape of the predictions, two examples are the Black-Scholes model and the SSVI model of [20]. The neural network is thus acting as a corrector of the prior model, enhancing its ability to reproduce observed market prices. The soft-constraints specification is guided by theoretical results from mathematical finance. The two elements together allow to build a realistic, flexible, and parsimonious model for the IVS. We benchmark our method against standard models and study its performance both on training and testing sets, as well as on synthetic data, and real market prices of contracts on the S&P 500 index. Numerical experiments suggest that our method appropriately captures the features of standard option pricing models. We show that increasing model capacity generally leads to better fits, and that the soft constraints generally help decreasing the fitting error and the convergence speed. Similar results are obtained when applying our method to real data, where an ablation study shows that constrained learning helps producing better volatility surfaces, both in interpolation and extrapolation.

**Novelty and significance.** Since options and related derivatives are actively exchange traded contracts that are used for risk-management and investment purposes, their fair valuation requires a reliable model for the IVS. The main ambition of this work is bridging the existing gap between a traditional challenge in finance and recent developments from the deep learning community, resulting in a trust-worthy, computationally efficient option pricing framework. This work is the first peer-reviewed published work to use soft-constraints to guide the training of a deep NN model for the IVS. Previous work used hardwired constraints which limits tremendously the neural network flexibility [14], or used constraints on the option price surface which requires additional transformations at each training step [44]. This work also appears to be the first suggesting a hybrid model for the IVS combining a neural network component, and a standard model component.

**Paper structure.** Section 2 reviews background knowledge such as the implied volatility surface, the no-arbitrage conditions, and formulates the modeling problem. We describe the methodology in Section 3. The numerical experiments and the empirical analysis can be found in Section 4. Section 5 discusses the broader impact. Section 5 concludes. More information on standard option pricing models, on implied volatility models, on the experiments, as well as additional results can be found in the online Appendix.

## 2 Background and objectives

### 2.1 The implied volatility surface

Let $\pi(K, \tau)$ denotes the market price of a call option with time to maturity $\tau > 0$ and strike price $K \geq 0$. Without loss of generality, we assume that the initial stock price is $S$, and that the interest-rate $r$ and dividend yield $\delta$ are constants. Denote by $C(\cdot)$ the Black-Scholes formula, namely

$$C(S, \sigma, r, \delta, K, \tau) = S^{-\delta\tau}\Phi(d_+) - \mathrm{e}^{-r\tau}K\Phi(d_-), \tag{1}$$

where $d_\pm = (\log(S/K) + (r-\delta)\tau)/(\sigma\sqrt{\tau}) \pm (1/2)\sigma\sqrt{\tau}$. More details on this formula can be found in Appendix A.2. The main objective of this paper is modeling the implied volatility whose definition is given below.

**Definition 2.1** *The* implied volatility $\sigma(k, \tau) > 0$ *is given by the equation*

$$\pi(K, \tau) = C(S, \sigma(k, \tau), r, \delta, K, \tau), \tag{2}$$

*with the (forward) log moneyness* $k = \log(K/Se^{(r-\delta)\tau})$. *The* implied volatility surface *is given by* $\sigma(k, \tau)$ *for* $k \in \mathbb{R}$ *and* $\tau > 0$.

**Interpreting the IVS shape.** For a fixed $\tau > 0$, $\sigma(k, \tau)$ with $k \in \mathbb{R}$ defines a volatility smile. If the smile has a *U shape*, then the tail of the log return $\log(S_T/S_t)$ distribution are thicker than the tails of the Gaussian distribution, and vice versa. If the smile exhibits a skew, then one side of the log return distribution is thicker than the other. For example, if the left side of a smile, which is a slice of the surface for a fixed $\tau$, is steeper than the right side, then the log price is more likely to experience large losses than large gains. The implied volatility surface provides a snapshot representation of valid option prices at a given time point. Although option prices fluctuate significantly over time, the shape and level of the implied volatility surface is fairly stable and large movements indicate important changes in market conditions.

### 2.2 Aribtrage-free surface

A static arbitrage is a static trading strategy that has a value that is both zero initially and always greater than or equal to zero afterwards, and a non-zero probability of having a strictly positive value in the future. In other words, an arbitrage costs nothing to implement while only providing upside potential, that is, it represents a risk-free investment after accounting for transaction costs. Under the assumption that economic agents are rational, any such opportunity should be instantaneously exploited until the market is arbitrage free. Therefore, option pricing models are designed in such a way that their call price surface $\pi(K, T)$ offers no possibility to implement such a strategy. Standard static arbitrage opportunities are described in Appendix A.3.

The absence of arbitrage translates into *constraints* on the call price surface $\pi(K, T)$, which in turn can be expressed as conditions that the implied volatility surface $\sigma(k, \tau)$ must satisfy [48, 20]. To express those conditions, we define the *total variance* of $\sigma(k, \tau)$,

$$\omega(k, \tau) = \sigma^2(k, \tau)\tau. \tag{3}$$

**Proposition 2.2** *Roper [48, Theorem 2.9] Let $S > 0$, $r = \delta = 0$, and $\omega : \mathbb{R} \times [0, \infty) \mapsto \mathbb{R}$. Let $\omega$ satisfy the following conditions:*

- *C1) (Positivity) for every $k \in \mathbb{R}$ and $\tau > 0$, $\omega(k, \tau) > 0$.*
- *C2) (Value at maturity) for every $k \in \mathbb{R}$, $\omega(k, 0) = 0$.*
- *C3) (Smoothness) for every $\tau > 0$, $\omega(\cdot, \tau)$ is twice differentiable.*
- *C4) (Monotonicity in $\tau$) for every $k \in \mathbb{R}$, $\omega(k, \cdot)$ is non-decreasing, $\ell_{\mathrm{cal}}(k, \tau) = \partial_\tau \omega(k, \tau) \geq 0$, where we have written $\partial_\tau$ for $\partial/\partial\tau$.*
- *C5) (Durrleman's Condition) for every $\tau > 0$ and $k \in \mathbb{R}$,*

$$\ell_{\mathrm{but}}(k, \tau) = \left(1 - \frac{k\,\partial_k \omega(k, \tau)}{2\omega(k, \tau)}\right)^2 - \frac{\partial_k \omega(k, \tau)}{4}\left(\frac{1}{\omega(k, \tau)} + \frac{1}{4}\right) + \frac{\partial_{kk}^2 \omega(k, \tau)}{2} \geq 0,$$

*where we have written $\partial_k$ for $\partial/\partial k$ and $\partial_{kk}$ for $\partial^2/(\partial k \partial k)$*
- *C6) (Large moneyness behaviour) for every $\tau > 0$, $\sigma^2(k, \tau)$ is linear for $k \to \pm\infty$.*

*Then, the resulting call price surface is free of static arbitrage.*

C1) and C2) are necessary conditions that any sensible model must satisfy. As for C3), it is merely sufficient to prove an absence of arbitrage when C4), C5), and C6) are also satisfied. Note that, assuming C3), C4) (respectively C5) and C6)) is satisfied if and only if the call price surface is free of calendar spread (butterfly) arbitrage [20]. C6) could be refined by imposing that $\sigma^2(k, \tau)/|k| < 2$ when $k \to \pm\infty$ to guarantee the existence of higher order implied moments, see [40, 7].

**Remark 2.3** *The Proposition 2.2 is derived under the assumption that $r = \delta = 0$ without loss of generality. Indeed, the static no-arbitrage constraints have the same functional forms as in C4)–C5)–C6) with non-zero parameters and the forward log moneyness $k$ defined in Definition 2.1.*

### 2.3   Problem formulation

**Data availability**. In practice, market data may need to be validated. This could be for a variety of reasons. First, exchange traded securities have at least two quotes, a bid and a ask price, and there is no guarantee that the average prices are arbitrage-free. Second, the observed prices may not be refreshed and thus not actionable, which may translate into notable input data noise. In addition, market data is typically sparse away from the money and dense close to the money. Indeed, far out of the money options are less likely to be exercised and are thus less likely to be used by their buyers. There is typically more demand and supply for contracts that are around the money.

**Modeling objectives**. The goal is to construct an IVS model $\sigma(k, \tau)$ that (I) generates options prices that are in line with market data, (II) is free of static arbitrage opportunities in the sense of Proposition 2.2, and (III) generalizes to unobserved data regions in a controlled fashion.

## 3   Methodology

### 3.1   Model and loss function

**Explanatory and target variables.** At a given time we observe triplets $(\sigma_i, k_i, \tau_i) \in \mathcal{I}_0$ where $\sigma_i$ is the market implied volatility (the target/response), and $(k_i, \tau_i)$ are the log moneyness and the time to maturity (the features/explanatory variables). In addition, we complement the sample with synthetic pairs $(k_i, \tau_i) \in \mathcal{I}_{\mathrm{C45}} \cup \mathcal{I}_{\mathrm{C6}}$ used to control the arbitrage opportunities and the model asymptotic behavior (see Section 3.2).

**Implied volatility model.** Our model for the total variance and implied volatility is given by

$$\omega_\theta(k, \tau) = \omega_{\mathrm{nn}}(k, \tau; \theta_1) \times \omega_{\mathrm{prior}}(k, \tau; \theta_2) \quad \text{and} \quad \sigma_\theta(k, \tau) = \sqrt{\omega_\theta(k, \tau)/\tau} \qquad (4)$$

for the parameters $\theta = \{\theta_1, \theta_2\}$, where $\omega_{\mathrm{nn}}$ and $\omega_{\mathrm{prior}}$ are the NN and prior models described below.

**Prior model.** $\omega_{\mathrm{prior}} : \mathbb{R}^2 \mapsto \mathbb{R}$ is a prior model with parameters $\theta_2$. An implied volatility model without prior is obtained by setting $\omega_{\mathrm{prior}} \equiv 1$ so that $\omega_\theta \equiv \omega_{\mathrm{nn}}$. The prior model choice is useful to ensure that the model generalization is compliant with a prescribed preferred behavior. Essentially, the BS prior is a parameter-free model matching the implied volatility's at-the-money (ATM) term-structure. As for the SSVI prior, it improves on the BS model by capturing both the smile and the skew of the surface. Both models are described in Appendix B.1 and B.2 respectively.

**NN model.** $\omega_{\mathrm{nn}} : \mathbb{R}^2 \mapsto \mathbb{R}$ is a standard feedforward multilayer neural network, namely

$$\omega_{\mathrm{nn}}(k, \tau; \theta_1) = \overset{n+1}{\underset{i=1}{\bigcirc}} f_i^{W_i, b_i}(k, \tau) \text{ with } f_i(x) = \begin{cases} g_i(W_i\, x + b_i) & i < n+1 \\ \alpha\, (1 + \tanh(W_{n+1}\, x + b_{n+1})) & i = n+1 \end{cases} \quad (5)$$

with $g_i$ an activation function, $\theta_1 = \{W_1, b_1, W_2, b_2, \ldots, \alpha\}$ the set of weight matrices and bias vectors, $\alpha$ a scaling parameter letting $\omega_{\mathrm{nn}}$ take values in $[0, \alpha]$, and $n$ the number of hidden layers. The functional choice $1 + \tanh$ in the last layer is not critical, yet it appeared to perform slightly better than the sigmoid function on a limited number of trials. Note that one can initialize the NN to take output value one, that is no NN correction, by setting $\alpha = 1$ and $W_{n+1} = b_{b+1} = 0$.

**Loss function.** We fit the network parameters and prior parameters $\theta$ by minimizing the loss function

$$\mathcal{L}(\theta) = \mathcal{L}_0(\theta) + \sum_{j=1}^{6} \lambda_j\, \mathcal{L}_{\mathrm{C}j}(\theta) \quad (6)$$

where the term $\mathcal{L}_0(\theta)$ is a prediction error cost, the terms $\mathcal{L}_{\mathrm{C}j}(\theta)$ for $j = 1, \ldots, 6$ materialize soft constraints aiming to ensure that the shape of $\{\omega_\theta(k, \tau);\ (k, \tau) \in \mathbb{R} \times \mathbb{R}_+\}$ is indeed a sensible implied volatility surface, and $\lambda_i$ for $j = 1, \ldots, 6$ are the corresponding penalty weights. Note that some parameters of the prior model may also be calibrated.

We let the prediction error be the sum of the root-mean-squared-error (RMSE) and the mean-absolute-percentage-error (MAPE),

$$\mathcal{L}_0(\theta) = \sqrt{1/|\mathcal{I}_0| \sum_{(\sigma_i, k_i, \tau_i) \in \mathcal{I}_0} (\sigma_i - \sigma_\theta(k_i, \tau_i))^2} + 1/|\mathcal{I}_0| \sum_{(\sigma_i, k_i, \tau_i) \in \mathcal{I}_0} |\sigma_i - \sigma_\theta(k_i, \tau_i)|/\sigma_i,$$

so as to penalize both absolute and relative errors. The IV values far out of the money can take values significantly larger than at the money. We found that the above loss function results in general a balanced allocation of prediction errors across the different moneynesses.

**Remark 3.1 (Soft versus hard constraints)** *An alternative approach to impose shape constraints on the mapping $\omega_\theta$ is to hard-wire them into the neural network architecture, as in [14] for example. However, hard constraints are difficult to impose on multilayer neural networks, may reduce the neural network's flexibility, and may lead to more challenging leaning routines, see [46].*

### 3.2 No-arbitrage conditions and synthetic grid

We explain how each of the constraints/conditions in Proposition 2.2 can be handled either by refining the architecture of the neural network, or by adding a penalty term to the loss function (6). Note that the conditions **C1)–C2)** are in principle satisfied by design of the NN. The mapping $\omega_\theta$ is twice differentiable as long as the activation functions $g_i$ and $g_{n+1}$ (as well as the prior model) are twice differentiable, in which case **C3)** is satisfied. An example of valid activation function is the SoftPlus given by $\ln(1 + \exp(x))$[1]. Hence we set $\mathcal{L}_{\mathrm{C}1} \equiv \mathcal{L}_{\mathrm{C}2} \equiv \mathcal{L}_{\mathrm{C}3} \equiv 0$.

As for the other three constraints, we control for

- calendar arbitrages with $\mathcal{L}_{\mathrm{C}4}(\theta) = 1/|\mathcal{I}_{\mathrm{C}45}| \sum_{(k_i, \tau_i) \in \mathcal{I}_{\mathrm{C}45}} \max(0, -\ell_{\mathrm{cal}}(k_i, \tau_i))$,
- butterfly arbitrages with $\mathcal{L}_{\mathrm{C}5}(\theta) = 1/|\mathcal{I}_{\mathrm{C}45}| \sum_{(k_i, \tau_i) \in \mathcal{I}_{\mathrm{C}45}} \max(0, -\ell_{\mathrm{but}}(k_i, \tau_i))$,
- and the asymptotic behavior with $\mathcal{L}_{\mathrm{C}6}(\theta) = 1/|\mathcal{I}_{\mathrm{C}6}| \sum_{(k_i, \tau_i) \in \mathcal{I}_{\mathrm{C}6}} \left| \partial^2 \omega_\theta(k_i, \tau_i)/\partial k \partial k \right|$,

where

$$\mathcal{I}_{\mathrm{C}45} = \left\{ (k, \tau) : k \in \left\{ x^3 : x \in \left[ -(-2k_{\min})^{1/3}, (2k_{\max})^{1/3} \right]_{100} \right\}, \tau \in \mathcal{T} \right\}$$

$$\mathcal{I}_{\mathrm{C}6} = \left\{ (k, \tau) : k \in \left\{ 6k_{\min}, 4k_{\min}, 4k_{\max}, 6k_{\max} \right\}, \tau \in \mathcal{T} \right\}$$

$$\mathcal{T} = \left\{ \exp(x) : x \in \left[ \log(1/365), \max(\log(\tau_{\max} + 1)) \right]_{100} \right\}$$

$k_{\min} = \min(\mathcal{I}_0^k)$, $k_{\max} = \max(\mathcal{I}_0^k)$, $\tau_{\max} = \max(\mathcal{I}_0^\tau)$, $[a, b]_x$ indicates an equidistant set of $x$ points between $a$ and $b$, and $\mathcal{I}_0^\tau$ and $\mathcal{I}_0^k$ are the sets of unique time to maturity and forward log

moneyness in $\mathcal{I}_0$. Note that it should always be that $\min(\mathcal{I}_0^k) < 0$ and $\max(\mathcal{I}_0^k) > 0$. The motivation for the above transformations is to obtain a denser grid around the money and for short maturities. The particular parametric choices for the above sets do not appear critical as long as they are sufficiently dense and cover the regions of interest.

Note that the loss $\mathcal{L}_{\mathrm{C6}}$ guarantees that $\omega$ is asymptotically linear and should in principle be refined to control the coefficient value to be less than 2, as discussed in [40]. However $\mathcal{L}_{\mathrm{C6}}$ has better training performance and proved to be compliant with C6) in our post training verifications. Furthermore, we only work with finite moneyness in practice and imposing an asymptotic condition on large but finite moneyness values may unnecessarily constraint the model.

### 3.3  Model training

The training procedure and the parameters choice are described in details in Appendix C.1.

## 4  Results

The first part presents results on synthetic data where it is easier to study and compare model predictions. The second part presents results on real-world data to modeling implied volatility surfaces extracted from S&P500 options prices. The synthetic and market data are described in Appendix C.2 and C.3 respectively. We always set $\lambda_4 = \lambda_5 = \lambda_6 = \lambda$ for some $\lambda \geq 0$.

### 4.1  Numerical experiments

**Smoothing behavior.** Figure 1 displays the trained model on synthetic data with a strong penalty $\lambda = 10$. A dot indicates the positioning $(k, \tau)$ of an observation used for training on figures in the top row. The figures on the first and second rows show that $\sigma_{\mathrm{prior}}$ does not succeed to reproduce the training data, while $\sigma_\theta$ does so perfectly. The figures in the last row display $\sigma_{\mathrm{prior}}$ and $\sigma_\theta$ on an extended grid of log moneyness, and also shows the NN correction before the scaling by $\alpha$ (center figure). As $\sigma_{\mathrm{prior}}$ fits at the money (ATM) term structure, $\sigma_\theta$ makes little correction to it on the region $k \approx 0$. On the other hand, $\sigma_\theta$ makes important corrections to $\sigma_{\mathrm{prior}}$ for out of the money options.

The same figure for additional scenarios ($\lambda$ value, prior model, and data limits) can be found in Appendix D.1. These additional results show that the model generalization is significantly better with a SSVI prior and with soft-constraints. Indeed, the IVS tends to flatten for large log moneyness values with a BS prior, whereas the present data is V shaped. Also, it is shown that the absence of soft-constraints can translate into non-realistic and possibly absurd out of sample predictions.

**Losses and convergence.**

Figure 2 displays statistics on the losses in (6) for trained models with different number of layers, neurons per layer, and penalty value $\lambda$. A total of 50 models with different random seeds for parameter initialization have been trained for 5'000 epochs for each configuration. Recall that, with $\lambda = x$, the weight on the soft constraints is $x$ times larger than the loss on the predictions errors. We see that increasing the number of layers or of neurons per layers generally decreases the RMSE and MAPE. We also see that the arbitrage opportunities vanish as $\lambda$ increases. The models prediction errors typically worsen as $\lambda$ increases, but the effect appears stronger for the lesser flexible models. We also observe that, when the number of layers is smaller, increasing the neurons per layer without enforcing more strictly the constraints can generate more severe arbitrage opportunities. But such opportunities disappear completely when the absence of arbitrage penalty is given a higher weight. Additional results can be found in Appendix D.2 for the BS prior where, in particular, we observe that the prediction errors are worse for this choice of prior model.

**Additional results.** In Appendix D.3 we show that the NN model suggested in [14] cannot capture the IVS on its full range of maturities and moneynesses for the synthetic data. We also illustrate the impact of increasing arbitrage opportunities on the losses in Appendix D.4.

### 4.2  Empirical results

**Smoothing market data.** Figure 3 displays S&P500 options data and trained model predictions with a strong penalty $\lambda = 10$ for the April 13-th 2018. This market data contains several thousands observations, and the corresponding IVS has a fairly complex shape compared to the previous synthetic data. The first two rows highlight that $\sigma_{\mathrm{prior}}$ fails to reproduce the market data, but that

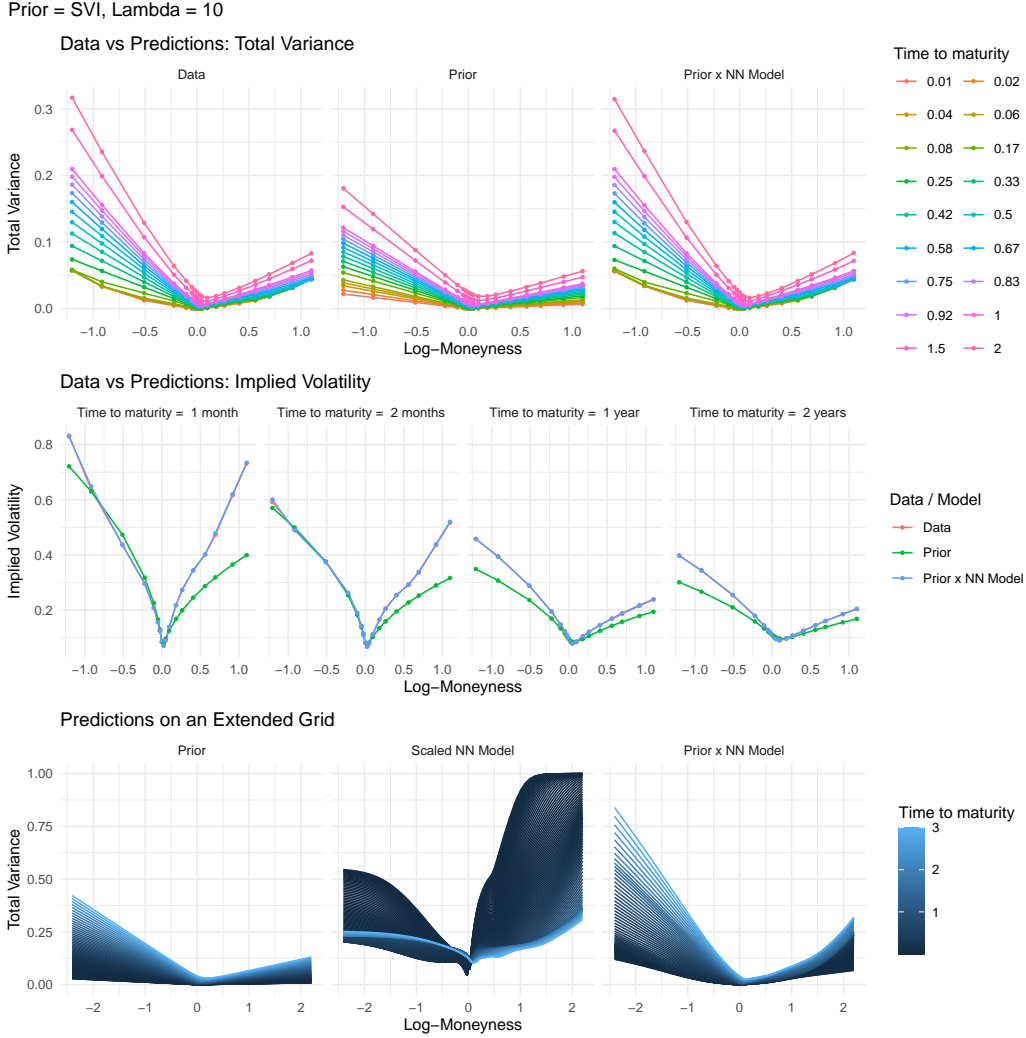

Figure 1: Synthetic data and trained model predictions for a specific configuration (scenario 12).

$\sigma_\theta$ is able to make accurate predictions. Notice that, because of data noise, the target IVS seem not to be arbitrage-free as some total variance slices cross each others, see upper-left figure. Yet, the predictions generated by $\sigma_\theta$ are perfectly smooth and the slices do not cross. The same figure for additional scenarios can be found in Appendix D.5.

**Backtests.** We perform the following exercise for each day in the sample. First, we split the daily sample into a training and a testing set. Second, we fit the model on the training set and evaluate its performance on the testing set. We use two different configurations for training and testing. In the interpolation setting, for each maturity, we randomly select half of the contracts. As such, we also sample options that are far out or in the money for training, and the testing error represents the approximation error for the range of moneyness that are actually observed. In the extrapolation setting, for each maturity, we select half of the contracts whose log moneyness is between the 10% and 90% of the log moneyness in the corresponding slice. This second filter therefore contains more observations around the money. Thus, we do not select options that are far out or in the money for training, and the testing error measures how well our model extrapolates. Finally, we again use three values for $\lambda$ in order to study how the arbitrage-related penalties affect the results.

In Table 1 we present our results for model trained daily on the S&P500 options data between January and April 2018, and for the subprime crisis period between September and December 2008, with a SSVI prior. First, we describe the RMSE and MAPE. As expected, training errors are generally below testing errors. Increasing the no-arbitrage penalty $\lambda$ leads to worse RMSE and MAPE metrics on the training set. However, larger $\lambda$ also implies similar or even less RMSE and MAPE metrics on

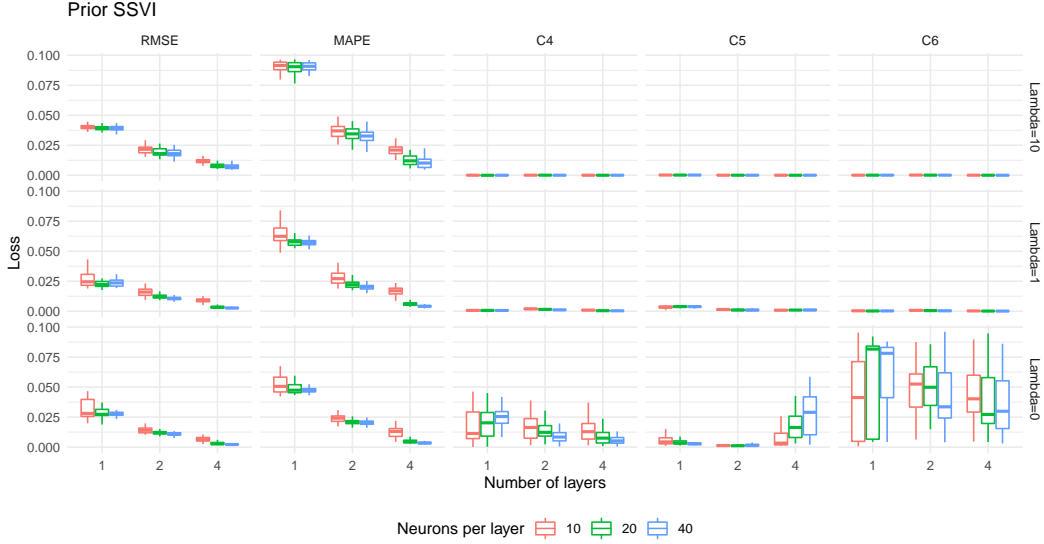

Figure 2: Losses for different number of layers, neurons per layer, and penalty value.

Table 1: Backtesting results for the SSVI prior (quantiles in %, Jan-Apr 2018 / Sep-Dec 2008)

| | | Interpolation | | | | | | Extrapolation | | | | | |
| | | Train | | | Test | | | Train | | | Test | | |
| Loss | $\lambda$ | $q_{05}$ | $q_{50}$ | $q_{95}$ | $q_{05}$ | $q_{50}$ | $q_{95}$ | $q_{05}$ | $q_{50}$ | $q_{95}$ | $q_{05}$ | $q_{50}$ | $q_{95}$ |
|---|---|---|---|---|---|---|---|---|---|---|---|---|---|
| RMSE | 10 | 0.4 / 0.7 | 0.5 / 3.1 | 1.3 / 19.9 | 0.4 / 0.9 | 0.5 / 3.3 | 1.4 / 18.8 | 0.2 / 0.3 | 0.3 / 1.1 | 0.4 / 3.5 | 2.2 / 2.7 | 5.0 / 6.3 | 8.0 / 12.0 |
| | 1 | 0.3 / 3.0 | 0.4 / 6.8 | 1.0 / 13.9 | 0.4 / 2.9 | 0.5 / 7.1 | 1.3 / 13.5 | 0.2 / 1.9 | 0.2 / 4.4 | 0.4 / 10.6 | 3.7 / 5.0 | 6.6 / 10.6 | 11.7 / 20.5 |
| | 0 | 0.2 / 0.4 | 0.3 / 1.3 | 0.5 / 4.0 | 0.3 / 1.6 | 0.5 / 3.5 | 4.8 / 10.8 | 0.2 / 0.3 | 0.2 / 0.9 | 0.3 / 3.3 | 3.1 / 5.5 | 7.5 / 11.6 | 18.1 / 26.7 |
| MAPE | 10 | 0.5 / 0.9 | 0.7 / 2.1 | 1.2 / 17.4 | 0.5 / 1.1 | 0.8 / 2.4 | 1.2 / 18.5 | 0.4 / 0.5 | 0.6 / 0.9 | 0.9 / 2.1 | 1.2 / 2.4 | 1.7 / 3.3 | 2.4 / 6.5 |
| | 1 | 0.5 / 4.0 | 0.6 / 8.2 | 1.2 / 12.2 | 0.5 / 4.3 | 0.7 / 8.1 | 1.3 / 12.9 | 0.3 / 2.2 | 0.5 / 6.3 | 0.9 / 11.0 | 1.5 / 5.6 | 2.3 / 9.7 | 3.3 / 13.1 |
| | 0 | 0.4 / 0.5 | 0.6 / 1.0 | 0.9 / 1.8 | 0.5 / 1.2 | 0.7 / 1.9 | 0.9 / 3.2 | 0.3 / 0.3 | 0.5 / 0.7 | 0.8 / 1.6 | 1.5 / 4.3 | 2.2 / 7.4 | 4.8 / 14.3 |
| C4 | 10 | 0.0 / 0.0 | 0.0 / 0.0 | 0.0 / 0.2 | 0.0 / 0.0 | 0.0 / 0.0 | 0.0 / 0.5 | 0.0 / 0.0 | 0.0 / 0.0 | 0.0 / 0.0 | 0.0 / 0.0 | 0.0 / 0.0 | 0.0 / 14.1 |
| | 1 | 0.0 / 0.0 | 0.0 / 0.0 | 0.2 / 0.1 | 0.0 / 0.0 | 0.0 / 0.0 | 0.2 / 0.1 | 0.0 / 0.0 | 0.0 / 0.0 | 0.0 / 0.1 | 0.0 / 0.0 | 0.0 / 0.0 | 0.1 / 0.1 |
| | 0 | 1.6 / 1.5 | 18.0 / 9.8 | 99+ / 99+ | 1.6 / 1.2 | 40.7 / 12.1 | 99+ / 99+ | 0.3 / 1.1 | 2.4 / 3.7 | 44.2 / 83.7 | 0.4 / 1.0 | 2.8 / 4.3 | 46.7 / 30.4 |
| C5 | 10 | 0.0 / 0.0 | 0.0 / 0.0 | 0.0 / 0.1 | 0.0 / 0.0 | 0.0 / 0.0 | 0.0 / 0.1 | 0.0 / 0.0 | 0.0 / 0.0 | 0.0 / 0.0 | 0.0 / 0.0 | 0.0 / 0.0 | 0.0 / 1.2 |
| | 1 | 0.0 / 0.0 | 0.0 / 0.0 | 0.1 / 0.0 | 0.0 / 0.0 | 0.0 / 0.0 | 0.5 / 0.0 | 0.0 / 0.0 | 0.0 / 0.0 | 0.0 / 0.0 | 0.0 / 0.0 | 0.0 / 0.0 | 0.5 / 0.0 |
| | 0 | 2.6 / 99+ | 72.1 / 99+ | 99+ / 99+ | 2.6 / 99+ | 63.8 / 99+ | 99+ / 99+ | 2.8 / 99+ | 24.2 / 99+ | 99+ / 99+ | 1.7 / 99+ | 19.0 / 99+ | 99+ / 99+ |
| C6 | 10 | 0.0 / 0.0 | 0.0 / 0.0 | 0.0 / 0.2 | 0.0 / 0.0 | 0.0 / 0.0 | 0.0 / 0.6 | 0.0 / 0.0 | 0.0 / 0.0 | 0.0 / 0.0 | 0.0 / 0.0 | 0.0 / 0.0 | 0.0 / 44.3 |
| | 1 | 0.0 / 0.0 | 0.0 / 0.0 | 0.0 / 0.0 | 0.0 / 0.0 | 0.0 / 0.0 | 0.6 / 0.0 | 0.0 / 0.0 | 0.0 / 0.0 | 0.0 / 0.1 | 0.0 / 0.0 | 0.0 / 0.0 | 0.0 / 0.0 |
| | 0 | 0.8 / 0.0 | 48.2 / 1.7 | 99+ / 99+ | 0.8 / 0.0 | 78.4 / 0.4 | 99+ / 99+ | 0.3 / 0.1 | 12.0 / 7.5 | 99+ / 99+ | 0.0 / 0.0 | 2.4 / 0.0 | 99+ / 99+ |

the testing set. Note that the RMSE for testing set of the extrapolation appears to be large compared to the MAPE. The extrapolation errors are most of the time larger than interpolation errors. This suggest that the high RMSE value is likely to be caused by deep out of the money options with large IV values. Regarding C4)–C6), we see that the models resulting from $\lambda = 1, 10$ are essentially arbitrage-free. As for the model resulting from no enforcement of the constraints (i.e., $\lambda = 0$), it generates arbitrage opportunities both in interpolation and extrapolation. Note that 99+ indicates a value larger than 99 which can happen for arbitrage losses when $\lambda = 0$.

Additional results for the BS prior, as well as for baselines models (Bates and SSVI models), can be found in Appendix D.6. Our approach surpasses the baseline models in terms of prediction accuracy, which should not be a surprise. Indeed, in Section 4.1 we showed that $\sigma_\theta$ could reproduce the IVS from a typical Bates model, hence it is likely to be at least as flexible. In addition, $\sigma_\theta$ with a SSVI prior $\sigma_{\text{prior}}$ is likely to perform better than the underlying baseline SSVI model.

**Additional results.** In Appendix D.7 we display the risk-neutral density and the local volatility corresponding to the scenario 12. The experience on Apple options in Appendix D.8 shows that our approach can fit complex shape surfaces, such as W-shaped smiles. We trained a model on the absolute IV errors weighted by the IV spread in Appendix D.9 which appears to converge faster but fails to learn the IVS shape far out of the money. The computational times for the empirical experiments are reported in Appendix D.10 and appear satisfying despite the experimental nature of the implementation.

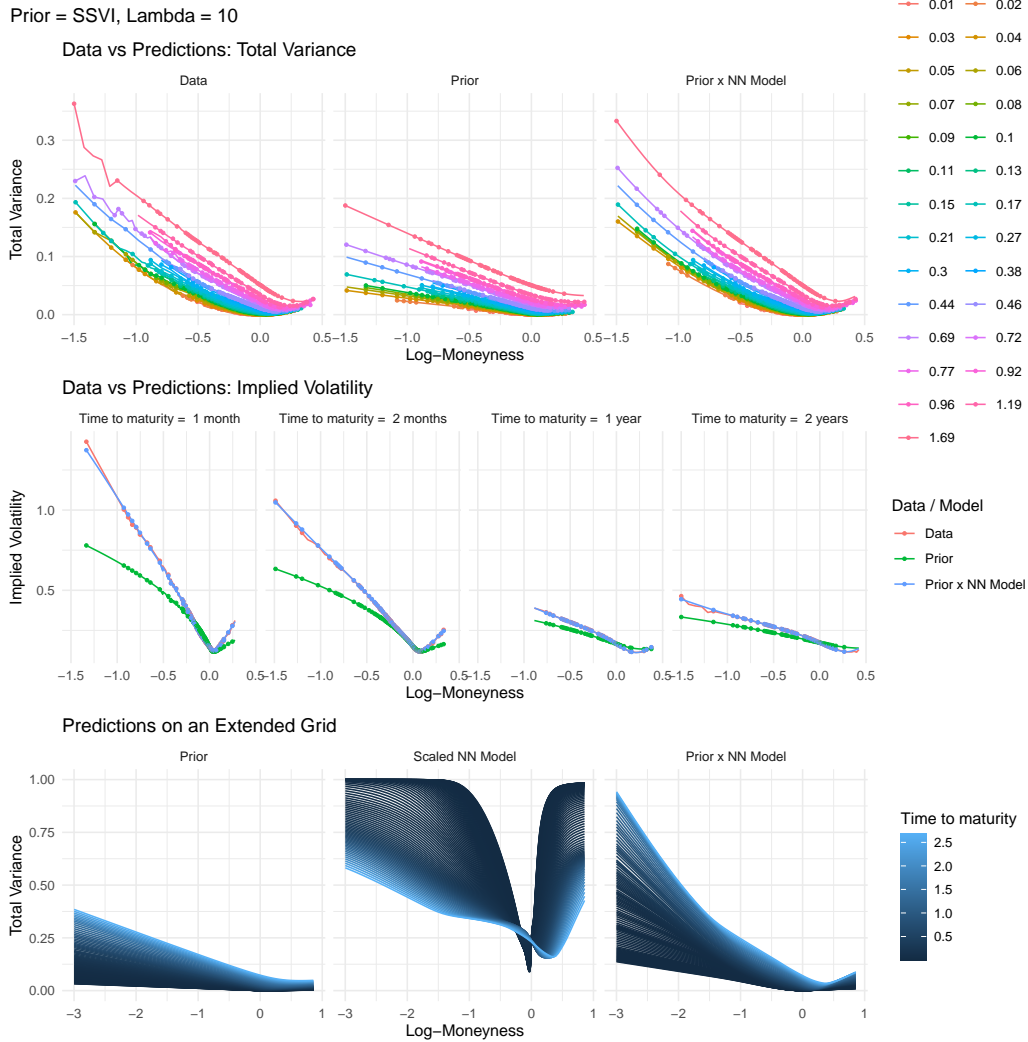

Figure 3: Market data and trained model predictions for a specific configuration (scenario 12).

## 5   Conclusion

We described a flexible methodology the price financial derivatives in an economically sensible way. This is achieved by modeling the implied volatility surface with a multilayer neural network and shaping it by penalizing the loss. We validate our approach with various numerical and empirical applications. The presented approach could be used as a building block to construct arbitrage-free models for multiple stocks, and for the IVS dynamics, as further discussed in Appendix E. Additionally, the loss function could be improved by avoiding to penalize models for predictions inside the spread, or using vega-weighting, as is commonly done by practitioners.

## Broader Impact

Financial markets play a central role in our economy, and our work could be used to generalize in a robust way the information available to participants. Options are actively traded securities that can be used for multiple reasons such as protecting pension portfolios against future losses, hedging against future fluctuations in crop prices for agricultural farmers, or speculation for hedge funds. Options are also used by economist and financial expert to extract market implied sentiment measures such as the VIX "fear" index. The financial risk created by options is often held by financial intermediaries, such as banks or brokers, which need appropriate tools to monitor and control their financial risk.

## Acknowledgments and Disclosure of Funding

The authors would like to thank Michael Roper for providing detailed comments and suggestions, as well as Serge Kassibrakis, Charles-Albert Lehalle, Johannes Wiesel, and participants at the 2019 SIAM conference on Financial Mathematics and Engineering in Toronto for helpful discussions.

The statements and opinions expressed in this article are those of the authors and do not represent the views of UBS (and/or any branches) and/or their affiliates.

## Footnotes

[1] We conjecture that one could also use ReLU activation functions with adjusted constraint conditions.

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
