[Supplementary Material]

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

# Online Appendix

## A    Background on option pricing models

### A.1    Literature review

The assumption of constant volatility in the Black-Scholes-Merton model has long been challenged empirically and various stochastic volatility models have been developed to tackle its limitations. Some examples are the Hull-White [36], the Stein-Stein [50], the Heston [32], the Variance-Gamma [45], the normal inverse Gaussian [4], the CGMY [10], the 4/2 [23], the Jacobi [3], rough Heston [15], and affine Volterra [2] models.

Albeit the development of more flexible models for stock prices, their statistical flexibility remained limited and they may be computationally too costly to calibrate for some real-world applications. For these reasons, parametric and nonparametric approaches have been developed aiming to interpolate, and sometimes to extrapolate, the implied volatility surface. These approaches includes the stochastic volatility inspired (SVI) and surface extension (SSVI) of [18, 20], and the smoothing spline techniques of [16, 13], among many others.

Several shallow neural networks approaches have also been developed to smooth option prices directly. [14] constructed a one hidden layer neural network monotonic or convex in its input coordinate, and taking only positive values. However, the construction is specific rendering it impossible to extend to multilayer neural networks, it performs poorly with both short and long maturities, and do not prevent all forms of static arbitrage opportunities. Recently, this model has been extended in a PhD thesis [53] by adding a gated unit layer linking the input to multiple models à la Dugas. [44] proposes a one hidden layer approach to model the implied total variance and his approach includes multiple ad-hoc rules. For examples, the extrapolation behavior to unobserved areas of the implied volatility surface is controlled for by adding discretionary data points, the training procedure is restarted until 25 neural nets are found to be arbitrage-free at selected strikes and maturities, the final implied volatility surface is obtained by aggregating over the best three models, and so on. The sigmoid-based approach of [38] to model the implied volatility smile is closely related to a neural network approach.

On a broader note, the financial applications of neural networks are booming as a consequence of the progress made in deep learning and of the availability of specialized software and hardware. They have for examples been used in [42, 43] to speed-up the pricing and calibration of options in stochastic volatility models, and in [8] to approximate optimal but intractable option hedging strategies with market frictions.

### A.2    The Black-Scholes (BS) formula

In the BS model, the dynamics of the stock price $S_t$ under the risk-neutral measure is given by

$$dS_t = (r - \delta)S_t dt + \sigma S_t dW_t \tag{7}$$

for some constants $r \in \mathbb{R}$, $\delta \geq 0$, and $\sigma > 0$, and where $W_t$ is a standard Brownian motion. Let $V_t$ denotes the price of a derivative at time $t$, then it satisfies the following partial differential equation,

$$0 = \frac{\partial V}{\partial t} + \frac{1}{2}\sigma^2 S^2 \frac{\partial^2 V}{\partial S^2} + (r - \delta)S\frac{\partial V}{\partial S} - rV. \tag{8}$$

Consider the call option payoff $(x - K)^+$ with strike $K$ and maturity $T$. Solving the PDE (8) with boundary condition $V_T = (S_T - K)^+$ with $S_t = S$ gives the following formula for the time-$t$ call option price $C$,

$$C(S, \sigma, r, \delta, K, \tau) = S^{-\delta\tau}\Phi(d_+) - e^{-r\tau}K\Phi(d_-), \tag{9}$$

where $\tau = T - t$, $d_\pm = (\log(S/K) + (r - \delta)\tau)/(\sigma\sqrt{\tau}) \pm (1/2)\sigma\sqrt{\tau}$ and $\Phi$ is the standard Gaussian CDF.

The dynamics of stock prices in the real world do not follow a geometric Brownian motion. Empirically validated stylized fact of stock log returns are, for examples, stochastic volatility and leverage effect which are not capture by (7). Despite its shortcomings, the BS model remains extremely popular in practice for its simple pricing formula, and the modeling complexity is moved to the input volatility parameter $\sigma$. Hence, if one understands the model and its limitations, the BS formula can be used as a Rosetta Stone to analyze market prices.

Figure 4: Payoffs of the calendar spread and butterfly as a function of the underlying asset price. For the calendar spread, $T_2 - T_1 = 1$, $K = 100$, and $\sigma = 0.25$. For the butterfly, $K_1 = 90$, $K_2 = 110$, and $K_3 = 100$.

### A.3 Static arbitrage opportunities

One can show that $\pi(K, T)$ is arbitrage-free if and only if it is free of calendar spread arbitrage and each time slice if free of butterfly arbitrage. A *calendar spread* is a strategy where one buys a call with a given maturity $T_1$ and sells another call with maturity $T_2$, both using the same strike, and where $T_1 > T_2$. At $T_1$, the value of the short call is $-\max(S_{T_1} - K, 0)$, whereas that of the long call, $\pi(K, T_2 - T_1)$, is always greater. A *butterfly* is a strategy where one buys two calls with strikes $K_1 < K_2$, and sells two other calls with strike $K_3 = (K_1 + K_2)/2$, but the same maturity. In Figure 4, we show the payoffs for each of the two strategies at $T_1$. Since the payoffs are always positives, they must have a nonzero initial price, for the market would otherwise allow for arbitrage opportunities.

### A.4 Financial engineering

The implied volatility surface plays a central role in the financial engineering toolbox. Thanks to automatic differentiation, the model in (4) can also be used to derive the local volatility surface as well as the risk neutral density, for examples. We refer to [19, Chapter 1] for more background.

**Risk neutral density.** Options are sometimes used to extract forward looking market sentiment indicators. For examples, the VIX and the SKEW indices[2] derive from the S&P500's mean, variance, and skewness as implied by option prices. In the present framework, we can reconstruct the entire fitted stock price density $p(x, \tau)$ at any future time $\tau$,

$$p(x, \tau) = \left.\frac{\partial^2 C}{\partial K^2}\right|_{K=x} = e^{r\tau} \Phi(d_+) \left.\frac{\ell_{\text{but}}(k, \tau)}{\sqrt{\omega(k, \tau)}}\right|_{K=x}.$$

**Local volatility.** The pricing of exotic and path-dependent options is often carried out using stochastic models equipped with a deterministic functional component that must be calibrated. This is the so-called local stochastic volatility function $\sigma_{\text{LV}}(k, \tau)$ with log moneyness $k$ which is given by,

$$\sigma_{\text{LV}}(k, \tau)^2 = \frac{\frac{\partial C}{\partial T} + (r - \delta)K\frac{\partial C}{\partial K} + \delta C}{\frac{1}{2}K^2\frac{\partial^2 C}{\partial K^2}} = \frac{\frac{\partial w}{\partial T}}{1 - \frac{k}{w}\frac{\partial w}{\partial k} + \frac{1}{4}\left(-\frac{1}{4} + \frac{1}{w} + \frac{k^2}{w^2}\right)\left(\frac{\partial w}{\partial k}\right)^2 + \frac{1}{2}\frac{\partial^2 w}{\partial k^2}}.$$

## B  Prior and baseline models

### B.1  Black-Scholes (BS) model

The BS model is the simplest possible prior model. In its most standard form, one would define $\omega_{\text{prior}}^{\text{bs}}(k, \tau) = \sigma^2\tau$ for some volatility $\sigma$. But, empirically, the at-the-money (ATM) total variance is not learn in $\tau$, and its term-structure can be inferred directly from market prices (see Appendix C). As a result, we redefine the BS prior as

$$\omega_{\text{prior}}^{\text{bs}}(k, \tau) = w_{\text{atm}}(\tau)$$

where $w_{\text{atm}}$ is described in Appendix C. Hence, this prior can be inferred directly from market prices by interpolating/extrapolating the ATM total variance.

## B.2 Surface Stochastic Volatility Inspired (SSVI)

The SVI parametrization for a volatility slice (single maturity) has been extended to the entire surface (SSVI) in [20]. We implemented the following version with a power-law parameterization of the function $\phi$

$$\omega_{\text{prior}}^{\text{ssvi}}(k, \tau) = \frac{w_{\text{atm}}(\tau)}{2}\left(1 + \rho\,\phi(w_{\text{atm}}(\tau))k + \sqrt{(\phi(w_{\text{atm}}(\tau))k + \rho)^2 + 1 - \rho^2}\right)$$

$$\phi(x) = \frac{\eta}{x^\gamma(1+x)^{1-\gamma}}$$

for some parameters $\rho \in (-1, 1)$, $\lambda \in (0, 1)$, $\eta > 0$, and where $w_{\text{atm}}$ is the at the money term-structure of the IVS as described in Appendix C. We also implemented the Heston-like parametrization for $\phi$ but the results were worse and are therefore not included. A generalization of the SSVI parametrization is given in [24].

## B.3 Stochastic volatility (SV) model

The stock price dynamics in the Bates model [5] is given by

$$dS_t/S_t = (r - \delta)dt + \sqrt{V_t}dW_t^1 + dN_t$$

$$dV_t = \kappa(\theta - V_t)dt + \sigma\sqrt{V_t}dW_t^2$$

where $r$ is the interest rate, $\delta$ is the dividend yield, $V_t$ is the spot volatility, $\theta$ is the long-run volatility, $\kappa$ is the speed of mean-reversion, $\sigma$ is the volatility of volatility, and $W_t^1$ and $W_t^2$ are two correlated Brownian motion with parameter $\rho$. The process $N_t$ is a compound Poisson process with intensity $\lambda$ and independent jumps $J$ with

$$\ln(1 + J) \sim \mathcal{N}\left(\ln(1 + \beta) - \frac{1}{2}\alpha^2, \alpha^2\right)$$

where the parameters $\alpha$ and $\beta$ determine the distribution of the jumps, and the Poisson process is assumed to be independent of the Brownian motions.

As the characteristic function of the log-price is known, we used the Fast Fourier transform method [11] in order to compute option prices efficiently.

**Calibration details.** SV models are typically calibrated on options prices since the corresponding implied volatility is not readily available (and would thus require an additional numerical procedure at each iteration). Note that, for this reason, it is not possible to use SV models as prior models.

We denote here $\pi_j$, $\sigma_j$, and $\nu_j$ the $j$-th option price, implied volatility, and Vega. Similarly $\hat{\pi}_j$ and $\hat{\sigma}_j$ denote the model option price and implied volatility. We calibrate the models by minimizing the Vega-weighted root-mean-square-error (RMSE)

$$\sqrt{\frac{1}{N}\sum_{j=1}^{N}\left(\frac{\pi_j - \hat{\pi}_j}{\nu_j}\right)^2} \tag{10}$$

where $N$ is the number of out-of-the-money options on a particular day. and where the Vega option Greek is given by and by

$$\nu = Se^{-\delta\tau}\phi(d_+)\sqrt{\tau} = Ke^{-r\tau}\phi(d_-)\sqrt{\tau}$$

for both Calls and Puts, where $\Phi$ and $\phi$ denotes respectively the standard Gaussian CDF and PDF.

The loss (10) is a computationally efficient approximation for the implied volatility surface RMSE criterion which follows by observing that

$$\sigma_j - \hat{\sigma}_j \approx \frac{\pi_j - \hat{\pi}_j}{\nu_j} \quad \text{when} \quad \pi_j \approx \hat{\pi}_j.$$

# C  Additional information on model training and data

## C.1  Training and model parameters

**Default parameters.** Unless stated otherwise, the NN will have 4 layers and 40 neurons per layer, the loss penalty values are $\lambda_4 = \lambda_5 = \lambda_6 = 10$, and the prior model is the SSVI. We chose this

configuration because it proved to be flexible enough to reproduce many model-based IVS, while always remaining arbitrage-free thanks to the large $\lambda$ penalty value.

**Parameters initialization.** We initialize the parameters so that the signal propagated through the layers do not explode or vanish, as motivated in [22]. The parameters $W_i$ and $b_i$ are all initialized by Gaussian random variables with mean zero and standard deviation $(n_{i-1}^{\mathrm{r}} + n_i^{\mathrm{r}})^{-1/2}$ where we recall that $n_i^{\mathrm{r}}$ denotes the output size of layer $i$.

**Optimization routine.** The total loss $\mathcal{L}(\theta)$ is minimized with the Adam optimizer [39]. As adaptive learning rate and early stopping have shown to significantly improve training [28, 25, 51, 12, 47, 33], we follow this approach. Starting with a learning rate of $0.01$, we let it decrease by a factor 2 on plateaus of length $500$ epochs when the total loss was not improved by more than $1\%$. The learning routine stops if $\mathcal{L}(\theta)$ has not improved by $1\%$ over $2'000$ epochs, and restarts using the initial learning rate until $4'000$ total epochs have been reached or until the total loss (6) is below $0.25\%$.

Note that the number of epochs is large compared to many deep learning applications, yet the training takes at most few minutes as the training data is also small (at most a few thousand samples at most and only two features). We are not using minibatch. In addition, although it takes many epochs for a randomly initialized model to converge to a good solution, we observed that training a model on new data using a previously trained but different model drastically improves the performance so that convergence can be achieved within seconds. We attribute this fact to the soft constraints which shape the neural network in complex ways that is hardly achievable with random initialization.

**At the money (ATM) total variance.** Because it can be inferred directly from market prices, we consider the ATM total variance term structure $w_{\mathrm{atm}}$ a model input as it feeds into the prior models. Indeed, we expect the prior model to be a first-order approximation of the surface and therefore to at least reproduce ATM values. This is especially important given that calls/puts close close to ATM are generally the most liquid.

We extract $w_{\mathrm{atm}}$ from the market data as follows. We know that it must be a positive and increasing function of $\tau$ given C1) and C4). For each maturities $\tau$ we collect the total variance $\sigma^2 \tau$ values corresponding to the contract closest to $k = 0$. We then use `interp_regular_1d_grid` from `TensorFlow Probability` if this term-structure is increasing. Because, for some dates, it was empirically not the case, we use the `SCAM` package to fit a spline monotonically increasing spline in $\tau$ with 10 knots and a smoothness penalty being selected by minimizing the generalized cross-validation criterion. Interpolations and extrapolations of the splines model on a fine grid are then used as constants and fed into `interp_regular_1d_grid` from `TensorFlow Probability`.

As both prior and NN models are trained, we observed that they sometimes compensate each others around the money. This implies that the ATM prior predictions sometimes deviate from $w_{\mathrm{atm}}$. While this is not an issue, we prefer if $\omega_{\mathrm{prior}}$ gives the best possible fit and let the NN compensate for its limitations. Therefore, we added an optional loss function $\mathcal{L}_{\mathrm{atm}}$ which encourage the NN model to be close to one for ATM values,

$$\mathcal{L}_{\mathrm{atm}}(\theta) = \frac{1}{|\mathcal{I}_{\mathrm{atm}}|} \left( \sum_{(k_i, \tau_i) \in \mathcal{I}_{\mathrm{atm}}} (1 - \omega_{\mathrm{nn}}(k_i, \tau_i; \theta_1))^2 \right)^{1/2}$$

for an ATM grid of points $\mathcal{I}_{\mathrm{atm}}$ given by $\mathcal{I}_{\mathrm{atm}} = \{(0, \tau) : \tau \in \mathcal{T}\}$. We always use a small penalty value of $\lambda_{\mathrm{atm}} = 0.1$ for this loss.

**Feature engineering.** Feature engineering is not used for the experiments reported in this paper, yet previous results showed that adding some features guided by expert judgment may lead to improved performance for a fixed model capacity. We observed that including feature variables inversely proportional to the time to maturity allowed to calibrate NN models with fewer layers and neurons for a given accuracy level. These features would take the form of $k\tau^{-\gamma}$ for $\gamma \in (0, 1)$ so that C1)–C2) remain satisfied and that the total variance remains asymptotically linear in $k$. We conjecture that this is because the implied volatility surface tends to sharply increase with $|k|$ at short horizons while being more flat at longer horizons.

**Code and hardware.** The method was implemented using tensorflow [1] and the experiments ran on Tesla K80 GPUs via Amazon Web Services. The code and data allowing to reproduce the numerical experiment with the synthetic data is provided in the supplementary. Because the real data was provided by a private provider, it unfortunately cannot be made publicly available.

## C.2 Model based (synthetic) data

To study the properties of our approach in a controlled setting, we create a synthetic dataset using Bates model, see Appendix B.3. We use the following grid $\mathcal{I}_0^{k,\tau}$ of log moneynss and maturities,

$$\mathcal{I}_0^{k,\tau} = \big\{(k,\tau) \,:\, k \in \mathcal{K}_0,\, \tau \in \mathcal{T}_0\big\}$$

with

$$\mathcal{K}_0 = \big\{\log(x) \,:\, x \in \{0.3, 0.4, 0.6, 0.8, 0.9, 0.95, 0.975, 1,$$
$$1.025, 1.05, 1.1, 1.2, 1.3, 1.5, 1.75, 2, 2.5, 3\}\big\}$$
$$\mathcal{T}_0 = \big\{i/52 \,:\, i \in \{0.5, 1, 2, 3\}\big\} \cup \big\{i/12 \,:\, i \in \{1, 2, \ldots, 11, 12, 18, 24\}\big\}$$

or in plain words for $\mathcal{T}_0$: half a week, one, two and three weeks, one to twelve months, eighteen months and two years.

The drift and diffusion parameters in the Bates model are $V_0 = 0.10^2$, $\theta = 0.25^2$, $\rho = -0.75$, $\kappa = 0.5$, and $\sigma = 1$. and the jump parameters are $\lambda = 0.1$, $\beta = -0.05$, and $\alpha = 0.15$. We also set the interest rate and dividend yield to zero.

## C.3 Market data

We use data European option price quotes on the S&P500 from the CBOE and provided by *OptionMetrics IvyDB US database* through the *Wharton Research Data Services*. Note that the quotes represent the best 15:59EST bid and ask and we use mid-quotes. We extracted implied volatility values for the periods September-December 2008 and January-April 2018 in multiple steps.

We first estimated the implied risk-free rate and dividend yield values. We started by computing the option mid prices by averaging the bid and ask prices, and use it as the reference prices henceforth. For each date $t$ and each contract maturity $T$, we use the options around the money ($\pm\,7.5\%$) and estimate coefficient in the linear regression

$$\mathrm{call}(K,t,T) - \mathrm{put}(K,t,T) = S_t\,\beta_{t,T}^S + K\beta_{t,T}^K$$

as guided by the put-call parity relation, where $S_t$ is the time-$t$ closing price of the index, also provided by *OptionMetrics*.

We then derive the maturity specific implied risk-free rate $r_{t,T} = -\log(\beta_{t,T}^K)/(T-t)$ and dividend yield $\delta_{t,T} = -\log(\beta_{t,T}^S)/(T-t)$. This allows us to compute the maturity specific log forward moneyness defined by $k = \ln(K/S_t) - (r_{t,T} - \delta_{t,T})(T-t)$ for each option, as well as the implied volatility values. Finally, Brent's method is used to extract the implied volatility on all options in the dataset.

We then apply a set of rules to select a realistic range of option contracts. We select only out of the money options, that is call options with $k > 0$ and put options with $k < 0$. We select contracts with time to maturity $(T-t)$ between 2 and 730 days, absolute log forward moyness $|k|$ less than 1.5, and implied volatility less than 300%. Table 2 provide daily summary statistics.

## D   Additional results

### D.1   Smoothing behavior

Figures 5 to 7 display the synthetic data and trained model predictions for the different scenarios summarized in Table 3. We observe that the predictions are typically worse with the BS prior than with the SSVI prior. Also, the predictions outside the observe data region (extrapolations) may exhibit strange behavior when $\lambda = 0$ so that the arbitrage opportunities are not controlled.

### D.2   Losses and convergence

Figure 8 displays the different losses for the BS prior using 50 random seeds and 5'000 epochs for each configuration. It highlights that the NN model does not succeed to compensate for the poor choice of prior model given the data.

### D.3   Comparison with Dugas et al.'s NN model

In [14] the authors proposed a one-layer neural network model to approximate the call, or put, price surface which is arbitrage-free by construction. We trained this model with 1'000 neurons for 50'000

Figure 5: Synthetic data and trained model total variance predictions.

Data vs Predictions: Implied Volatility

Figure 6: Synthetic data and trained model implied volatility predictions for selected maturities.

Figure 7: Trained prior and NN predictions for the synthetic data.

Table 2: Daily statistics for the implied volatility dataset.

|  |  | Min | Q1 | Median | Q3 | Max |
|---|---|---|---|---|---|---|
|  | contracts | 3946 | 4405 | 4788 | 4980 | 5291 |
|  | maturities | 32 | 33 | 34 | 34 | 35. |
|  | $k$ min | -1.5 | -1.5 | -1.5 | -1.5 | -1.4 |
| Jan-Apr/2018 | $k$ max | 0.3 | 0.4 | 0.4 | 0.4 | 0.5 |
|  | $\sigma_{IV}$ min | 0.0 | 0.1 | 0.1 | 0.1 | 0.1 |
|  | $\sigma_{IV}$ max | 1.6 | 2.4 | 2.8 | 2.9 | 3.0 |
|  | contracts | 513 | 631 | 748 | 790 | 873 |
|  | maturities | 12 | 12 | 13 | 13 | 14 |
|  | $k$ min | -1.5 | -1.5 | -1.3 | -1.1 | -0.8 |
| Sep-Dec/2008 | $k$ max | 0.6 | 0.8 | 1.0 | 1.1 | 1.2 |
|  | $\sigma_{IV}$ min | 0.2 | 0.2 | 0.2 | 0.3 | 0.3 |
|  | $\sigma_{IV}$ max | 0.7 | 1.3 | 1.7 | 2.2 | 3.0 |

Figure 8: Losses for different number of layers, neurons per layer, and penalty value

Table 3: Configurations for the different scenarios, the second column indicates the fraction of near the money data retained for training.

| prior | $k$ prop. | $\lambda_4$ | $\lambda_5$ | $\lambda_6$ | scenario |
|---|---|---|---|---|---|
| BS | 0.8 | 0 | 0 | 0 | 1 |
| BS | 0.8 | 1 | 1 | 1 | 2 |
| BS | 0.8 | 10 | 10 | 10 | 3 |
| BS | 1 | 0 | 0 | 0 | 4 |
| BS | 1 | 1 | 1 | 1 | 5 |
| BS | 1 | 10 | 10 | 10 | 6 |
| SSVI | 0.8 | 0 | 0 | 0 | 7 |
| SSVI | 0.8 | 1 | 1 | 1 | 8 |
| SSVI | 0.8 | 10 | 10 | 10 | 9 |
| SSVI | 1 | 0 | 0 | 0 | 10 |
| SSVI | 1 | 1 | 1 | 1 | 11 |
| SSVI | 1 | 10 | 10 | 10 | 12 |

Figure 9: IV surface for the Dugas et al. NN model trained on the synthetic data.

epochs on the put price surface from the Bates synthetic data, see Section C.2. Figure 9 displays the train and fitted implied volatility surfaces. We observe large IV errors for the Dugas et al. model, indeed some IV values are no even within the range $(0.01\%, 10'000\%)$ and are thus not displayed. This is because small price errors for far out of the money options translate into large implied volatility errors. In addition an option whose time value is smaller than zero is erroneous and cannot be mapped to an IV value, recall that the time value is defined as the difference between the option price and its intrinsic value $((K - S_0)^+$ here). Figure 10 highlights this issue for the put prices predicted by fitted the Dugas et al. model.

### D.4 Impact of arbitrage opportunities on the losses

We illustrate the impact of erroneous IVS data on the different losses by adding random Gaussian noise with volatility parameter $\eta$ to the synthetic log-IV data. That is, if $\sigma$ is an IV value then we multiply it by $\exp(\epsilon)$ where $\epsilon \sim \mathcal{N}(0, \eta)$. The first row of Figure 11 displays the perturbed total variance surfaces, and the second row different losses for the trained NN model with penalty value $\lambda \in \{0.01, 10\}$. We observe that a larger noise level translates into larger prediction errors for the $\lambda = 10$ model as the more severe arbitrage opportunities are smoothed away. The prediction errors

Figure 10: Put option time value for the Dugas et al. NN model trained on the synthetic data.

Figure 11: Noise impact on the IVS shape and trained NN losses.

for $\lambda = 0.01$ model also increases with $\eta$, although at a lower rate, because its capacity remains fixed while the surface complexity further increases.

### D.5   Smoothing real data

Figures 12 to 14 display the market data and trained model predictions for the different scenarios summarized in Table 3. The conclusions are similar as for the synthetic data. We observe that the choice of prior has a notable impact on the NN model, and that the NN modle is always significantly more accurate than its corresponding prior.

### D.6   Backtests

Table 4 displays the backtesting results with the Black-Scholes prior. Table 5 displays the backtesting results for the benchmark models (Bates and SSVI).

Figure 12: Market data and trained model total variance predictions.

Figure 13: Market data and trained model implied volatility predictions for selected maturities.

Figure 14: Trained prior and NN predictions for the market data.

Table 4: Backtesting results for the BS prior (quantiles in %, Jan-Apr/2018 - Sep-Dec/2008)

| | | Interpolation | | | | | | Extrapolation | | | | | |
|---|---|---|---|---|---|---|---|---|---|---|---|---|---|
| | | Train | | | Test | | | Train | | | Test | | |
| Loss | $\lambda$ | $q_{05}$ | $q_{50}$ | $q_{95}$ | $q_{05}$ | $q_{50}$ | $q_{95}$ | $q_{05}$ | $q_{50}$ | $q_{95}$ | $q_{05}$ | $q_{50}$ | $q_{95}$ |
| RMSE | 10 | 0.9 / 0.7 | 3.6 / 2.1 | 14.3 / 13.2 | 1.1 / 1.0 | 3.8 / 2.7 | 14.7 / 14.8 | 0.2 / 0.3 | 0.4 / 1.2 | 4.2 / 5.1 | 5.4 / 7.4 | 10.8 / 16.6 | 21.1 / 31.2 |
| | 1 | 0.5 / 0.7 | 3.0 / 18.3 | 14.4 / 39.1 | 0.6 / 1.4 | 2.9 / 18.9 | 14.9 / 44.4 | 0.2 / 0.3 | 0.3 / 1.1 | 4.4 / 3.3 | 6.0 / 9.0 | 11.5 / 16.6 | 21.6 / 29.4 |
| | 0 | 0.4 / 0.3 | 2.8 / 0.8 | 14.4 / 7.6 | 0.6 / 1.8 | 3.0 / 5.0 | 14.3 / 25.2 | 9.5 / 0.3 | 12.7 / 0.8 | 15.4 / 3.1 | 24.6 / 15.2 | 29.7 / 34.2 | 35.7 / 69.5 |
| MAPE | 10 | 0.9 / 1.1 | 1.2 / 1.8 | 4.3 / 7.7 | 0.9 / 1.3 | 1.3 / 2.0 | 4.4 / 6.9 | 0.5 / 0.5 | 0.8 / 1.0 | 2.3 / 4.5 | 2.5 / 3.8 | 3.7 / 5.5 | 8.3 / 12.0 |
| | 1 | 0.6 / 1.2 | 0.9 / 22.2 | 4.4 / 29.2 | 0.7 / 1.5 | 0.9 / 23.0 | 4.2 / 29.6 | 0.4 / 0.5 | 0.7 / 1.0 | 2.4 / 2.7 | 2.5 / 3.7 | 3.9 / 5.4 | 8.8 / 10.2 |
| | 0 | 0.6 / 0.5 | 0.8 / 0.8 | 4.0 / 1.5 | 0.6 / 1.4 | 0.9 / 2.2 | 4.0 / 4.3 | 24.2 / 0.4 | 31.3 / 0.8 | 40.4 / 1.7 | 33.3 / 7.1 | 40.9 / 18.5 | 50.7 / 34.4 |
| C4 | 10 | 0.0 / 0.0 | 0.0 / 0.0 | 0.1 / 0.3 | 0.0 / 0.0 | 0.0 / 0.0 | 0.1 / 0.3 | 0.0 / 0.0 | 0.0 / 0.0 | 0.0 / 0.0 | 0.0 / 0.0 | 0.3 / 0.0 | 1.4 / 0.9 |
| | 1 | 0.0 / 0.0 | 0.1 / 0.0 | 0.3 / 2.3 | 0.0 / 0.0 | 0.1 / 0.0 | 0.3 / 3.7 | 0.0 / 0.0 | 0.0 / 0.0 | 0.1 / 0.3 | 0.0 / 0.0 | 0.7 / 0.1 | 1.9 / 13.2 |
| | 0 | 0.8 / 12.1 | 11.1 / 96.4 | 75.2 / 99+ | 0.6 / 12.6 | 11.3 / 99+ | 39.4 / 99+ | 0.0 / 7.7 | 0.0 / 37.9 | 0.0 / 99+ | 0.0 / 10.0 | 0.0 / 27.8 | 0.0 / 99+ |
| C5 | 10 | 0.0 / 0.0 | 0.0 / 0.0 | 0.0 / 0.0 | 0.0 / 0.0 | 0.0 / 0.0 | 0.0 / 0.1 | 0.0 / 0.0 | 0.0 / 0.0 | 0.0 / 0.0 | 0.0 / 0.0 | 3.5 / 0.0 | 15.9 / 0.8 |
| | 1 | 0.0 / 0.0 | 0.0 / 0.0 | 0.1 / 7.5 | 0.0 / 0.0 | 0.0 / 0.0 | 0.1 / 99+ | 0.0 / 0.0 | 0.0 / 0.0 | 0.0 / 0.3 | 0.0 / 0.0 | 3.7 / 0.6 | 16.5 / 8.4 |
| | 0 | 1.2 / 91.8 | 20.0 / 99+ | 99+ / 99+ | 1.2 / 99+ | 21.5 / 99+ | 99+ / 99+ | 0.0 / 99+ | 0.0 / 99+ | 0.0 / 99+ | 0.0 / 99+ | 0.0 / 99+ | 0.0 / 99+ |
| C6 | 10 | 0.0 / 0.0 | 0.0 / 0.0 | 0.0 / 0.1 | 0.0 / 0.0 | 0.0 / 0.0 | 0.0 / 0.8 | 0.0 / 0.0 | 0.0 / 0.0 | 0.0 / 0.0 | 0.0 / 0.0 | 0.0 / 0.0 | 0.0 / 0.0 |
| | 1 | 0.0 / 0.0 | 0.0 / 0.0 | 0.0 / 0.2 | 0.0 / 0.0 | 0.0 / 0.0 | 0.1 / 33.5 | 0.0 / 0.0 | 0.0 / 0.0 | 0.0 / 0.0 | 0.0 / 0.0 | 0.0 / 0.0 | 0.0 / 4.1 |
| | 0 | 2.3 / 7.0 | 37.1 / 99+ | 99+ / 99+ | 2.0 / 10.7 | 33.9 / 99+ | 99+ / 99+ | 0.0 / 0.0 | 0.0 / 69.9 | 0.0 / 99+ | 0.0 / 0.0 | 0.0 / 0.0 | 0.0 / 99+ |

Table 5: Backtesting results for the benchmarks (quantiles in %, Jan-Apr/2018 - Sep-Dec/2008)

| | | Interpolation | | | | | | Extrapolation | | | | | |
|---|---|---|---|---|---|---|---|---|---|---|---|---|---|
| | | Train | | | Test | | | Train | | | Test | | |
| Benchmark | Loss | $q_{05}$ | $\mu$ | $q_{95}$ | $q_{05}$ | $\mu$ | $q_{95}$ | $q_{05}$ | $\mu$ | $q_{95}$ | $q_{05}$ | $\mu$ | $q_{95}$ |
| BATES | RMSE | 1.6 / 2.0 | 2.9 / 5.1 | 5.3 / 10.8 | 1.7 / 2.1 | 2.8 / 5.2 | 5.0 / 9.7 | 1.0 / 1.2 | 1.6 / 3.1 | 4.3 / 8.1 | 2.4 / 2.5 | 4.6 / 7.0 | 8.7 / 14.4 |
| | MAPE | 3.7 / 3.5 | 6.1 / 5.9 | 11.8 / 11.4 | 3.6 / 3.5 | 6.1 / 6.1 | 11.8 / 11.5 | 3.2 / 2.4 | 5.6 / 4.5 | 20.6 / 12.6 | 4.1 / 4.0 | 6.4 / 7.2 | 18.1 / 15.6 |
| SSVI | RMSE | 4.0 / 3.1 | 6.7 / 7.3 | 10.8 / 15.0 | 4.0 / 2.9 | 7.0 / 7.3 | 10.6 / 15.7 | 1.5 / 1.8 | 2.2 / 4.6 | 4.0 / 11.3 | 6.6 / 5.0 | 11.2 / 10.8 | 16.2 / 21.4 |
| | MAPE | 5.1 / 4.3 | 7.2 / 8.6 | 9.7 / 13.8 | 5.0 / 4.1 | 7.2 / 8.7 | 9.6 / 13.3 | 3.2 / 2.0 | 4.6 / 6.5 | 7.2 / 11.9 | 6.5 / 5.9 | 8.8 / 9.9 | 11.2 / 14.1 |

## D.7 Risk-neutral density and local volatility

We apply the approaches described in A.4 for a model fitted on the synthetic data of C.2. Figure 15 displays the price density (first row) and the local volatility (second row) with respect to the log moneyness and for different horizons.

## D.8 W-shaped smile

We extracted from *OptionMetrics* IVS database the implied volatility values for call options on Apple on the July 21st 2015. In this database the option prices have already been mapped into IV values for standardized strikes and maturities, that is we have 11 maturities and 17 strikes per maturities. The particularity of this selected surface is that it exhibits W-shaped IV smiles. This sometimes happens before scheduled news releases such as company earnings. These are challenging surfaces to fit for classical models. We calibrate our model with the SSVI prior on the entire surface. Figure 16 displays the fitted total variance surface, as well as the fitted smiles for the first two maturities for which we observe that the SSVI prior fails to capture the W-shaped smile whereas the NN model succeeds.

## D.9 Spread weighted errors

We train the NN model with a different loss function,

$$\mathcal{L}_0(\theta) = 1/|\mathcal{I}_0| \sum_{(\sigma_i, k_i, \tau_i) \in \mathcal{I}_0} \frac{|\sigma_i - \sigma_\theta(k_i, \tau_i)|}{1 + \sigma_i^{\text{ask}} - \sigma_i^{\text{bid}}}$$

where $\sigma_i^{\text{bid}}, \sigma_i^{\text{ask}}$ are the implied volatility corresponding to bid and ask option prices, and $\sigma_i$ corresponds to the option mid-price. Figure 17 displays the results for scenario 12. We observe that the predictions for far out of the money options are not as accurate as with the default loss function. This is not surprising as far out of the money options typically have large IV spread values.

## D.10 Computational time

Table 6 reports the summary statistics for the NN model training times. Note that the implementation producing the results reported in this paper calls Python TensorFlow from R, hence experiences overhead. In addition, the code itself was not written to be efficient, but rather for exploratory

Figure 15: Density function and local volatility for scenarios 6 and 12.

Figure 16: W-shaped smiles for the options on the Apple stock on July 21st 2015.

Figure 17: Market data and trained model predictions for a specific configuration (scenario 12) and trained on spread weighted errors.

Table 6: Computation times (in minutes)

| Prior | Period | Min | Mean | Sd | Max |
|---|---|---|---|---|---|
| BS | Sep-Dec/2008 | 1.0 | 1.3 | 0.6 | 4.7 |
| | Jan-Apr/2018 | 0.9 | 1.4 | 0.7 | 6.6 |
| SSVI | Sep-Dec/2008 | 1.8 | 2.1 | 0.9 | 8.0 |
| | Jan-Apr/2018 | 1.5 | 2.2 | 1.0 | 8.0 |

flexibility. In our experience, a simple TensorFlow implementation in Python would typically runs at least 2-4 times faster.

# E  Discussion

## E.1  Extensions

The presented approach focuses on a single stock and observations at a single time because this already represents a challenging and relevant practical problem. Still, we discuss how one could approach the modeling of multiple stocks, and the IVS at multiple time points.

**Multi-asset.** It is straightforward to extend the current approach to create a model for multiple IVSs. Indeed, one may consider a model taking as input an input categorical variable that will the volatility surface. This approach would allow to transfer reuse and transfer features learned from on surface to another, which could be highly beneficial in situations where few observations are available for IVS. This advantage typically comes at the cost of building and training a deeper and larger NN for a joint model than for individual models. Note that, additional no-arbitrage constraints may have to be imposed in the specific case of implied volatility surfaces for currency pairs.

**Multi-day.** Modeling the IVS at multiple time points could be useful to transfer information forward from the past, which may be allows to build a more resilient image of the IVS a given time point. This may be achieved in different ways. For example, the parameters $\theta$ of the previous model could be used as initial values for the model to be newly fitted. Alternatively, one may think of propagating forward in time a latent factor as performed in recurrent neural networks.

Another application could the construction of a generative model for future IVSs which could be used for risk-management, for example. Such a generative model would be required to simulate entire IVSs given present and past values. Our approach could be used in combination with a generative module to guarantee that the fake IVSs are sensible.