[Reviews · NeurIPS 2020]

Review 1

Summary and Contributions: This paper proposes a NN approach to predict implied volatility surfaces (IVSs). The proposed method can be combined with standard IVS models in quantitative finance, thus providing a NN-based correction when econometric models fail at replicating observed market prices.

Strengths: + The goal of this paper is clear, and the paper is generally well-written. + This paper aims at an important topic in quantitative finance: Accurately predict IVS. + Adding some non-arbitrage conditions into the loss function is novel. + The model presentation, mathematical formulas and symbol definitions are all very clear. + The is the first work that suggests the use of soft-constraints on the IVS values to guide the training of a deep model.

Weaknesses: - Lack of theoretical justification for the proposed method - A short period of data is used in the experiments, making the empirical results not so convincing. I'm not sure if the model still works for periods of important financial events, e.g., the financial crisis of 2007–2008. - The reference format is not consistent. - A typo in LINE 190 for the MAPE equation.

Correctness: Yes

Clarity: Yes

Relation to Prior Work: Yes

Reproducibility: Yes

Additional Feedback:


Review 2

Summary and Contributions: This work proposed a neural network approach to fit and predict implied volatility surfaces. The proposed approach met the non-arbitrage principle, which was a key in real finance applications. Their method was evaluated on synthetic data and SP500 data. Overall, it is a high-quality submission, but I do have the concern: the paper studied a very niche problem in finance industry, and many designs/concepts/novelties are heavily tied with the problem itself, so may have quite limited audience for a machine learning venue.

Strengths: * The writing was very clear and easy to follow, though it was not quite friendly for the non-finance background audience, because many materials were placed in appendix, probably due to the page limits. * I quite like the key design of the predictive model: instead of producing the total variance/implied volatility, the authors chose to use the NN to ‘correct’ an existing prediction from a mathematical finance model (e.g., SVI). What the NN model produced was a ‘rescale’ factor. This design was novel as far as I can tell, and it pointed out a neat way to build a hybrid model of NN and mathematical finance model.

Weaknesses: * "This work is the first to suggest the use of soft-constraints on the IVS values to guide the training of a deep model" was an over-claim. The use of soft constraints on NN models for fitting implied volatility surfaces have been investigated in Yu 2018 [60]. * The submission contains several typos, e.g., Line 33, ‘fir’ -> ‘for’, and Line 190: MAPE error formulation. * I can understand that the final layer activation is \alpha(1+tanh(.)) with \alpha>0 is chosen to guarantee the positivity, but (1+tanh(.)) is an odd choice, why not simply use sigmoid? * It was not very clear how condition (6), i.e., large moneyness behaviour is linked to the design of the loss term \mathcal{L}_{C6}. * For the choice of prior model, SVI doesn’t stand for the STOA mathematical finance models, to name a few, the authors may want to try SSVI, extended SSVI, rough volatility model.

Correctness: Yes

Clarity: Yes

Relation to Prior Work: Yes

Reproducibility: Yes

Additional Feedback: NB: broader impact was not presented


Review 3

Summary and Contributions: The authors propose an approach for modeling the implied volatility surface that works by using a neural network to correct the output of a prior (parametric) model, such as Black-Scholes. The main idea is a simple one: parametric models like BS can fail to reliably describe the volatility surface "away from the money" and may introduce spurious and undesirable arbitrage opportunities, but one can "correct" these artifacts using a secondary model (in this case a neural network). The paper is very well written and while many of the details are pushed into the appendix, they did a very good job of describing a complex problem - modeling the implied volatility surface - while maintaining focus on their contribution. My main concern is that this paper has a very narrow scope and may be better suited to a financial machine learning venue. In addition, the authors failed to include a section on broader impact.

Strengths: The primary strength of this paper is that the authors present a practical solution to an important financial modeling problem and clearly demonstrate an improvement over standard parametric approaches to modeling the volatility surface, in terms of modeling market data, providing a stable price surface away from observed data (tail behavior) and in enforcing the no arbitrage conditions. This benefit is clearly visible in the results shown in the main paper.

Weaknesses: I only see a few weaknesses in this paper, other than a lack of relevance to the broader NeurIPS community. First, why did the authors use a feed-forward NN model? With only 2 features I wonder if a linear model with an appropriate (say quadratic or cubic) basis expansion could provide enough modeling flexibility. Second, I would liked to have seen some of the training details make it into the main paper. I assume standard gradient based optimization methods were used, but some detail would be nice - especially in regards to the arbitrage constraints. In addition, some more detail on how the prior model was used to sample points to enforce the zero arbitrage condition and tail behavior is needed. Also, how large is the market data for a typical option used in the analysis? Finally, the authors mention a few existing applications of NNs to volatility surface smoothing in the related work section, but they don't compare to the models in the results section. The proposed system seems solid, but a more realistic baseline than BS or SVI would help drive this home.

Correctness: As mentioned above, my main criticism with the empirical results is that the authors didn't directly compare to existing NN-based smoothing methods. Its not important that the proposed method is significantly better in all cases, but providing a more realistic baseline would be helpful.

Clarity: The paper is very well written. Only have a few minor comments: Line 26: "...interpolation and extrapolation..." Line 33: "...flexible predictors for applications..." Line 108: "and that the interest rate" Figure 2 appears before Figure 1 in the pdf

Relation to Prior Work: yes.

Reproducibility: Yes

Additional Feedback:


Review 4

Summary and Contributions: This manuscript presents a neural network approach to fit, correct and extrapolate implied volatility surfaces from option prices. The two main novelties introduced are: 1/ The use of soft constraints (penalty terms) to characterize no arbitrage conditions (the "correct" and "extrapolate" part), as well as the neural network based fit with the inclusion of a model based prior. The product of this prior and the neural network fit produces the surface.

Strengths: This work is significant for a few main reasons, of which: - Classical fitting techniques come in two flavors: one relying on rigid models (typically just the prior component), which struggles to fit the data. Another relying on ad-hoc parametrizations that are more flexible, but are hard to calibrate not to include arbitrages (and are usually slower). This works tries to mix the best of both worlds - Dual to vol fitting is option pricing. The pricing function is highly non-linear, unknown in closed form (for american options) and requires involved numerical methods (e.g. PDE solver, or Longstaff-Schwarz Monte Carlo simulations). Having a robust and fast vol fitting procedure will minimize calls to the pricers, resulting in drastic speed ups

Weaknesses: - The authors do not compare their work in terms of accuracy and speed to other techniques - The objective function could use more work, especially given the data available to the authors as inputs - Some details on how the authors pre-process their inputs are unclear

Correctness: My only concern about the methodology is that it does not reflect the aim of the paper. The authors are trying to achieve two things: 1- fit the data (to the extent where the data is arbitrage free) and 2- correct any arbitrage in the data. The authors do report RMSE and MAPE for the fitting part, as well as different metrics for the arbitrage penalties C_{4,5,6}. However, it is unclear to me when: - RMSE/MAPE is large, is it because the fit doesn't go through the arb free data (so a real fitting issue), or is it because the data has arbs and the correction part is doing its job? One could try to guess what the answer is using C_{4,5,6} but this is not enough. For example, including metrics for how much the data itself is arb free would enlighten the reader as to whether a large RMSE is expected or not.

Clarity: The paper is well written, short and concise. The appendix is a great addition and goes straight to the point. No concerns on that end.

Relation to Prior Work: The introduction clearly details prior work in this field.

Reproducibility: Yes

Additional Feedback: \section*{Comments} \subsection*{General comments} From a theoretical standpoint, the two novelties that the authors claim to introduce -combining a prior model with a neural network and using soft constraints for the no arbitrage conditions, are of critical importance and make this manuscript worthy of a publication to me. From a practical standpoint, the implementation details -whether regarding the loss function, the neural network inputs or the training set, could use more work before producing a useful implied volatility surface superior to the ones produced by more classical techniques. A comparison with these techniques would have been most welcome. \subsection*{Objective function} \begin{itemize} \item $\mathcal{L}_{C_0}$: The authors weight every data point in an equal manner (in each component of the loss). They should at least weight those observations by: 1/The bid/ask spread (no need to fit perfectly when the market is wide, and focus on ATM options which are usually tighter than the wings) and 2/ The `vega' $\frac{\partial p}{\partial \sigma}$ of each options, which penalizes errors on options which are highly sensitive to $\sigma$ more than ones which are less sensitive to it. \item $\mathcal{L}_{C_0}$: Even better than the suggestion above: this loss term should be computed in `price' space and not in `volatility' space, only penalizing the $\ell^1$ and/or $\ell^2$ norm if the theoretical fitted price falls outside of the bid/ask range. \item $\mathcal{L}_{C_5}$: I wouldn't use a large value of $\lambda$ for enforcing the butterfly constraint as is: many other considerations have to be taken into account for such a constraints, mostly pertaining to bid/ask prices and transaction costs. Using an equivalent condition in price space might make it easier to write a more realistic version. \item $\mathcal{L}_{C_6}$: Regarding $\mathcal{I}_{C_6}$, and more precisely $\mathcal{K}_{C_6}$: First this condition is applied outside of $[4 \min(\mathcal{I}_0^k); 4 \max(\mathcal{I}_0^k)]$, which seems a bit extreme to me. If my understanding of $\mathcal{I}_0^k$ is correct, this no arb condition is applied for $|k| > 6$ only: I doubt there are any tradeable strikes in that region for maturities less than 3 months (most of the volume traded are for maturities less than 1 month). On the other hand, I'm not sure much `no arbitrage smoothing' is done for these extreme (but tradeable) strikes. A square root extrapolation only in regions where no strikes are traded isn't very useful/meaningful. My advice is to enforce $\mathcal{L}_{C_6}$ for strikes with $|z| > 8$, where $z = \frac{k}{\sqrt{w_{atm}}}$, and with a denser synthetic grid. \end{itemize} \subsection*{Data set} \begin{itemize} \item How are the implied volatilities $\sigma_i$ (and similarly the ATM volatility used in the Black-Scholes prior $w_{prior}$) obtained? This question is twofold: 1/ What pricer was used to imply the volatilities? In particular is it an American pricer (options on S\&P500 are mostly American in my knowledge)? Is it PDE based or Monte-Carlo based? In both cases, what are the implementation details? 2/ What root finding numerical algorithm was used to deduce implied volatilities, given the pricer? I think both questions should be answered by the authors. \item Data set: Table 2 provides some statistics about the data used, and seems to imply that only one quote (or rather pair of bid/ask quotes) per option, expiry and day was used. How were these quotes selected? For e.g. are they averaged throughout the day? Are they taken at the same time every day (if so when)? Same question for the index value used in the price to vol conversion, or in the dividend computation. \item Regarding the dividend computations, more details would be welcome: the authors explain how to deduce a dividend from put/call parity, but do not explain how they `average' the results over strikes for example. In particular, what do they do if there is high variance? \item Trying to fit the wings using the mid volatilities (the ones obtained by averaging bid and ask of puts on the lower wing, or calls on the upper wing) is, in general, a bad idea: the procedure will grossly underestimate that parameter in those regions, as well as the variance of the data there. My advice is to compute a log-moneyness dependent mid price, that would coincide with the mid price at $k=0$, and with the relevant side of the quote as $|k| \to \infty$ (or even better, use a $z = \frac{k}{\sqrt{w_{atm}}}$ weighted average instead of $k$). \item In the data selection procedure, the authors use a filtering rule of $|k| < 2$. I think using a constant $2$ might be a bit naive, given the wide variations of $k$ as a function of maturity. My advice is to use a variable $z = \frac{k}{\sqrt{w_{atm}}}$ that has a more stable scale, and more suited for such cutoffs. \end{itemize} \subsection*{Results} \begin{itemize} \item The authors might be limited by the number of samples in the dataset, but if possible I would like to see examples of implied vol plots and statistics on RMSE,MAPE for maturities less than 1 month: the implied volatility surface has a more complex structure there (especially at the wings), and this is where neural network fitting should shine compared with traditional ad-hoc parametrizations. \item Figure 4: I find the numbers in the interpolation region for $C_{4,5,6}$ and $\lambda = 0$ a bit high compared with RMSE and MAPE numbers, especially given that the market for S\&P500 options is as efficient as it can get. Any comments from the authors on that aspect -e.g. are there other considerations that are not captured by the no arbitrage conditions that explains that? \end{itemize} \section*{Further investigations} \begin{itemize} \item Most of the training set contains `U' shaped implied volatlities. Some data points should exhibit a `W' shape, with ATM volatility being a local \emph{maximum}. This happens when expiration is right after a major macro-economic event with binary outcomes (elections, FED announcement etc.), with agents taking sizeable bets on these outcomes. It would be interesting to see if the proposal captures this behavior, as the inputs can be used to decide which regime we are in (ATM convexity). \item The authors should try to experiment iterating the procedure, by starting with a Black-Scholes prior, and by selecting the prior at step $N$ to be the solution at step $N-1$. This should allow capturing the full complexity of the implied volatility surface, closer to maturity. \end{itemize} \section*{Typos and minor comments} \begin{itemize} \item l.108: `the in\textbf{s}terest rate' \item l.547: `indicate an\textbf{d} equidistant' \item Given that the authors seems to have settled for 4 layers and 40 neurons per layer (C.1),it would be nice if Figures 2 and 3 reflected this choice of parameters. \end{itemize}

[Author Response · NeurIPS 2020]

We thank the reviewers for their comments to improve the paper. The main contributions have been well identified, i.e.
using soft-constraints to control for arbitrage opportunities, and the NN-based correction of a quant finance-based prior.

**Broader impact and relevance to the NeurIPS community (R3/R4):** We agree that we study a specific problem, but
our primary subject area is "Applications: Quantitative Finance and Econometrics" and the concern applies to most
applications papers. For the finance industry, second only to tech as sponsor of NeurIPS2019, and in our opinion a
significant part of the community (maybe explaining the subject area's existence), the problem's relevance cannot be
overstated. Most banks and hedge-funds use IV surfaces (IVSs) and need such models. And derivatives are important
not only to the financial industry, but to any firm aiming at quantitatively manage risk via hedging, e.g., agricultural
commodities for farmers, fuel consumption for airlines, FX rates for companies having multi-currency activities, etc.
We will clarify by bringing Appendix E.2 (our current "broader impact section", not detailed enough) to the main text,
expand on it with such a discussion along with numbers relevant to derivatives markets and IVSs.
**Additional experiments (R1/R4/R5):** We tested, our approach works both in high-vol periods (e.g., 09/2008) and with
"W" shapes. We will add figures/tables to the appendix. We will also add computation times, an experiment with spread
weighting for $\mathcal{L}_0$, and we are open to suggestions for additional benchmarks beyond SVJ/Bates & SSVI (see Table 5).
**Training & data (R4/R5):** The information is in Appendix C because of the page limit, but we will add the key details
in the main paper, and expand the section on the real data pre-processing. Data is from optionmetrics (option/underlying
prices & $r_f$), historically recording the best 15:59EST bid/ask. We use midquotes everywhere, the put-call parity & $r_f$
to extract div yields, Brent's method to extract the IV, and CBOE's S&P500 options are European (i.e., no pricer).

**Reviewer #1**   **Theoretical justification:** Apologies, we do not understand. Economic theory states that put-call prices
should be arbitrage free; anomalies would otherwise be exploited until they disappeared. Thus, models like ours,
mathematically guaranteeing the absence of arbitrage, albeit via soft-constraints, are theoretically justified.
**Reviewer #3**   **Yu 2018 [60]:** Thanks for letting us know. We cited this PhD thesis for the gated NN ideas published in
[59] in which constraints on the price surface are used. But we were not aware of the work on IVS constraints, this part
of the thesis was not published. Albeit our work was done independently, we will correct the over-claim and give credit.
**Final layer:** Good remark, the "1 + " allows the NN to be initialized at 1 by setting $\alpha = 1$, $W_{n+1} = 0$, $b_{n+1} = 0$, i.e.
start from the prior. We tried the sigmoid, and tanh appeared to work better. We will discuss this in the manuscript.
**C6:** Apologies if we misunderstand, but if $\partial^2 \omega / \partial k \partial k = 0$ for large $k$, then $\omega$ is at most linear asymptotically.
**SVI not SOTA:** Agreed, we actually use the SSVI with power-law parametrization (see Appendix B.2), we will change
SVI into SSVI everywhere. The Heston-like parametrization gave worst results and thus was excluded. The eSSVI, i.e.
making $\rho$ a function of the ATM total variance, can be included easily (already implemented). But eSSVI and other SVI
extensions are, in our opinion, less popular in the industry. We will gladly add results for any SOTA method of interest.
**Reviewer #4**   **Feed-forward NN:** We are not sure whether you mean replacing our NN correction by a basis expansion-
based NN correction, or replacing the whole approach altogether by a basis expansion. The latter has been done with
cubic splines in [16]. But it doesn't allow to extrapolate, and the training algorithm requires a fixed grid and has
convergence issues when the grid is too dense (i.e., limited to IVSs with few options). For the former, it is surely
possible, albeit our final layer has some advantages (see answer to R3). For this application, what matters is mostly
model capacity and ease of computing partial derivatives to penalize the NN plugged on top of the prior via soft
constraints. And while both shallow and deep NNs can approximate continuous function with arbitrary precision, deep
NNs can reach the same accuracy as shallow ones with less neurons (see e.g., Thm 1, Foggio et al., PNAS 2020).
**Other NN baselines:** We implemented [13] but it didn't work well on the S&P500 data. We can include these results,
as well as any baseline of your choosing. Typically, NN approaches for the price surface struggle with deep OOM
options (small price errors become large IV errors). This is not the case with ours which controls the asymptotic IV.
**Reviewer #5**   **Ojective function:** Spread weighting for $\mathcal{L}_0$ is an excellent idea, overlooked for simplicity as it didn't
fundamentally change the results, but we can add a figure. In our experience, vega weighting helps to fit prices, where
small errors can lead to large IV errors. Also, not penalizing models for predictions within the spread is important. In
our opinion, this idea is similar to that for $\mathcal{L}_{C5}$; extremely relevant in the real world, somewhat beyond the scope of this
paper. Regarding $\mathcal{L}_{C6}$, its application in regions without tradeable strikes is extreme by design. Because $C6$ describes
the asymptotic behavior, we aimed to regularize/"clip" the NN when $|k| \to \infty$, hence the strikes much larger than
observed. We agree that one could be maturity dependent, i.e. use a $\tau$-dependent $\mathcal{K}_6^\tau$, and will update accordingly.
**RMSE/MAPE attribution:** It is difficult to disentangle the two cases in real data, especially given that we use midquote
IVs. We will add to the appendix and experiment where we created synthetic data with a model generating arbitrage
opportunities, applied our method, and compared metrics knowing the "true" values for $C_{4,5,6}$.
**Results:** Thanks for the comment on short maturities, we will add loss statistics as well as a figure to the appendix.
Note also that, in the figures about real data, the first maturity (0.02) corresponds to 7 days. We are computing arbitrage
losses on the grids $\mathcal{K}_{C45}$ and $\mathcal{K}_{C_6}$. On a grid point, unless it is "close" to a real option, nothing constraints the NN.
**Figures 2 & 3:** Thanks for noticing, we will update accordingly.
**Additional feedback:** We highly appreciate the efforts in providing detailed and insightful comments. We drafted
point by point detailed answers but unfortunately couldn't fit them on one page.

[Meta-Review · NeurIPS 2020]

We thank your for your submission. The reviewers felt that this paper presents an interesting problem for ML community and tackles it in a novel way. However, during the discussion period reviewers found that there are some major concerns with this paper: 1. The motivation is not clear. Volatility (sigma_i) is inherently unobserved quantity. However, the methodology presented here seems to rely on its availability. But if one has a mechanism to produce sigma_i's what is the need for this method? 2. Reproducibility. Experimental section (and appendix) seem to be missing a lot of details. We strongly encourage the authors to make the code (including data preprocessing part) open. Please refer to the reviews for details. Personally, I also found the MAPE+RMSE part of the objective poorly motivated. It felt like multiple things has been tried and this just seemed to work best. This being said, reviewers felt that novelty of the proposed problem and solution outweight this. Please take a serious effort to address these comments in your final version.